



# Landsat-based Irrigation Dataset (LANID): 30-m resolution maps of irrigation distribution, frequency, and change for the U.S., 1997-2017

Yanhua Xie[1,2], Holly K. Gibbs[1,2,3], Tyler J. Lark[1,2]

[1]Nelson Institute Center for Sustainability and the Global Environment (SAGE), University of Wisconsin-Madison, Madison, 53726, USA
[2]DOE Great Lakes Bioenergy Research Center, University of Wisconsin-Madison, Madison, 53726, USA
[3]Department of Geography, University of Wisconsin-Madison, Madison, 53706, USA

*Correspondence to*: Yanhua Xie (xie78@wisc.edu)

**Abstract.** Data on irrigation patterns and trends at field-level detail across broad extents is vital for assessing and managing limited water resources. Until recently, there has been a scarcity of comprehensive, consistent, and frequent irrigation maps for the U.S. Here we present the new Landsat-based Irrigation Dataset (LANID), which is comprised of 30-m resolution annual irrigation maps covering the conterminous U.S. (CONUS) for the period of 1997 – 2017.  The main dataset identifies the annual extent of irrigated croplands, pastureland, and hay for each year in the study period.  Derivative maps include layers on maximum irrigated extent, irrigation frequency and trends, and identification of formerly irrigated areas and intermittently irrigated lands. Temporal analysis reveals that 38.5 million hectares of croplands and pasture/hay have been irrigated, among which the yearly active area ranged from ~22.6 to 24.7 million hectares. The LANID products provide several improvements over other irrigation data including field-level details on irrigation change and frequency, an annual time step, and a collection of ~10,000 visually interpreted ground reference locations for the eastern U.S. where such data has been lacking.  Our maps demonstrated overall accuracy above 90 % across all years and regions, including in the more humid and challenging-to-map eastern U.S., marking a significant advancement over other products, whose accuracies ranged from 50 to 80 %. In terms of change detection, our maps yield per-pixel transition accuracy of 81 % and show good agreement with U.S. Department of Agriculture reports at both county and state levels. The described annual maps, derivative layers, and ground reference data provide users with unique opportunities to study local to nationwide trends, driving forces, and consequences of irrigation and encourage the further development and assessment of new approaches for improved mapping of irrigation especially in challenging areas like the eastern U.S. The annual LANID maps, derivative products, and ground reference data are available through https://doi.org/10.5281/zenodo.5003976 (Xie et al., 2021).

## 1 Introduction

Irrigated agriculture is vital to global food security.  Irrigation helps stabilize farm production by enhancing land productivity that would otherwise be lower due to water limitations to plant growth.  In the U.S., approximately 14.6 percent of the total cropland is irrigated (USDA-NASS, 2019). Despite this relatively small proportion, irrigated agriculture plays a significantly



disproportionate role in agriculture, accounting for major proportions of the economic value and environmental impacts; irrigated farms account for 54 percent of the total value of crop sales (USDA-NASS, 2021). However, agricultural irrigation uses over 40 percent of total freshwater withdrawals and 80 to 90 percent of consumptive water use in the U.S. (Dieter et al., 2018; USDA, 2019). As a result, improved management and understanding of irrigation use and trends offers a key leverage

point to improve the sustainability of U.S. agriculture.

Knowledge of the spatial and temporal patterns of irrigation is a crucial first step to improve this understanding and management, and to help policymakers make decisions to support sustainable water use for crop production. However, the spatiotemporal patterns of irrigation and their impacts are not well understood, even for data-rich countries like the U.S. This lack of data and quality hampers a much larger body of research and applications, such as the modelling of land surface

characteristics, climate and weather, and the growth of crops and other vegetation. For those applications that do incorporate irrigation modules, they are typically based on infrequently updated coarse-resolution global maps that cannot represent the precise locations of irrigated fields (Zaussinger et al., 2019; Ozdogan et al., 2010). As such, there is significant need for field-relevant resolution maps of irrigated agricultural land and its temporal changes. The value of such detailed irrigation information is further magnified as society formulates strategies towards sustainable use of limited water resources from local

to global scales under the context of increasing food and fuel demands, climate change and extremes, and population growth (Lark et al., 2015; Rosegrant et al., 2009; Seto et al., 2012; Seager et al., 2012; Mcdonald et al., 2011).

Despite the growing importance of field-level irrigation information to a wide array of research questions and applications, currently available irrigation maps covering the entire or part of the Conterminous United States (CONUS) suffer from limitations related to spatial resolution, update frequency, geographical coverage, and mapping accuracy (Table 1). For

example, the spatial resolution of all nationwide maps (except for LANID-US 2012) ranges from 250-m to kilometers, which is problematic for many local applications that require accurate field characterization (Wardlow and Callahan, 2014; Deines et al., 2017; Ozdogan and Gutman, 2008; Xie et al., 2019; Brown and Pervez, 2014). Just as importantly, all these nationwide irrigation maps are infrequently updated and mapped at either a single date or at intervals of five years to decades (e.g., Shrestha et al. (2021), Brown and Pervez (2014) and Ozdogan and Gutman (2008)). Due to annual crop rotations, fallow practices, and

climate variation, however, irrigation use and decision making are extremely dynamic. Accordingly, more timely information is needed to understand changes in irrigation and the associated impacts including water use and availability.

The recent years have witnessed an unprecedented development of land use/cover mapping owing to the increasing availability of high- to moderate-resolution remote sensing data and improvement of computing capacity (e.g., emergence of cloud computing platforms). While annual continental to global products of some land use/cover types have been created in a near

operational manner (e.g., forest, water, and urban) (Hansen et al., 2013; Pekel et al., 2016; Gong et al., 2020), frequent fine-scale irrigation mapping remains challenging due to the cryptic nature of the irrigation signal and the lack of ground reference data needed to train and validate machine learning and other classifiers. The data gaps are particularly problematic in the midwestern and eastern U.S., where more abundant water resources have led to less concern and monitoring of irrigated land use.



**Table 1. Currently available irrigation maps covering part to the entire CONUS. The boldfaced maps are compared with LANID in the Results section.**

| Products | Spatial coverage | Resolution | Update frequency | Citations |
|---|---|---|---|---|
| **Global Irrigated Area Map (GIAM)** | Global | 10 km rescaled to 1 km | Single map, 2000 | Thenkabail et al. (2009) |
| **Global Map of Irrigation Areas (GMIA)** | Global | 10 km | 5-year, 1995, 2000, and 2005 | Siebert et al. (2005) |
| Synthesized map of global irrigated area | Global | 1 km | Single map, covering 1999-2012 | Meier et al. (2018) |
| Global Food-Support Analysis Data (GFSAD) | Global | 1 km | Single map, 2010 | Teluguntla et al. (2015) |
| Global Land Cover Map (GlobCover) | Global | 300 m | Single map, 2009 | Esa (2015) |
| Global Land Cover Characteristics (GLCC) | Global | 1 km | Single map, 1992 | Loveland et al. (2000) |
| Global Rainfed, Irrigated and Paddy Croplands (GRIPC) | Global | 500 m | Single map, 2005 | Salmon et al. (2015) |
| **MODIS-based Irrigated Agriculture Dataset (MIrAD)** | CONUS | 250 m | 5-year interval, 2002-2017 | Pervez and Brown (2010) |
| **MODIS-based Irrigation Fraction (MIF)** | CONUS | 500 m | Single map, 2001 | Ozdogan and Gutman (2008) |
| **USDA-NASS Irrigation Statistics** | U.S. | County-level | 5-year interval, 1997-2017 | https://www.nass.usda.gov/AgCensus/index.php |
| USGS-verified irrigated lands | Western U.S. | Field | Vary across states, 2002-2017 | Brandt et al. (2021) |
| Landsat-based Irrigation Dataset 2012 (LANID 2012) | CONUS | 30-m | Single map, circa 2012 | Xie et al. (2019) |
| **Annual Irrigation Maps – High Plain Aquifer (AIM-HPA)** | High Plains Aquifer | 30-m | Annual, 1984-2017 | Deines et al. (2019) |
| **IrrMapper** | Western CONUS | 30-m | Annual, 1986-2018 | Ketchum et al. (2020) |

This paper presents the newly created annual 30-m resolution irrigation maps and their comparisons with existing products. The maps (named LANID – Landsat-based Irrigation Dataset) cover CONUS for the years between 1997 and 2017, which

were built upon a past effort of irrigation mapping for the year 2012 (Xie et al., 2019), with key improvements in training sample generation, classification design, and accuracy assessment (Xie and Lark, 2021). The maps presented here also include a newly mapped component – irrigated pasture and hay – that was not explicitly included in the preliminary version presented in Xie and Lark (2021). In addition to the LANID maps, we present the collected ground truth data, which is particularly important for irrigation mapping efforts that require such a dataset to train or validate machine learning algorithms, especially

where it has not been available in the humid eastern U.S. Additional products include maps of irrigation frequency, maximum extent, irrigation trends, formerly and intermittent irrigated areas. In the following sections, we briefly review the methods used to generate these data and then present our maps and their comparisons with existing products that cover the entire U.S.



## 2 Methods

Our new LANID product contains 21 annual maps that characterize irrigation status of croplands, pasture, and hay across
CONUS for the years from 1997 to 2017. We first created annual maps of irrigated croplands (i.e., LANID_V1) using a
supervised decision tree classification based on a novel training sample generation method and satellite-derived and
environmental variables (see details in Xie and Lark (2021)). Because LANID_V1 does not explicitly include irrigated pasture
and hay, which is an important component of total irrigation particularly in the western U.S., we addressed this limitation by
applying the same machine learning method but using different mask layers and areal reference for training sample generation
and classification (Fig. 1). The maximum extent of pasture and hay for the west was derived from the USGS National Land
Cover Database (NLCD) and USDA Cropland Data Layer (CDL), identifying pixels that had been classified as pasture/hay in
NLCD or non-alfalfa hay in CDL within any year between 1992 and 2017. To reduce competition between this pasture and
hay mask and the one used for irrigated cropland mapping, we removed those pixels that had been classified as irrigated
cropland in LANID_V1.  The county-level areal reference of irrigated pasture and hay was calculated as the deficit of
LANID_V1-based irrigated cropland area compared to USDA NASS reported area, which includes all types of irrigated
agriculture.

A key element of the LANID methodology is a novel way to generate training samples covering the entire country. To account
for climate difference and mapping complexity, CONUS was divided into western and eastern states based on a climatic
transition near the 100th meridian and training data were created corresponding to each region (Fig. 2). We used an automated
method to generate training samples for the western states. For the years when USDA-NASS county-level irrigation statistics
are available (i.e., 1997, 2002, 2007, 2012, and 2017), we adopted the thresholding method proposed by Xie et al. (2019) to
automate training sample generation, which assumed that irrigated lands appear greener than those that are rainfed. For non-
census years, optimal thresholds were estimated based on relationships of crop greenness between non-census and census
years. The calibrated and estimated thresholds were used to segment yearly maximum Landsat-based greenness index (GI)
and enhanced vegetation index (EVI) to derive two intermediate irrigation maps per year, which were overlaid to identify
consistent classification as potential training samples. As a result, the generated potential training samples were evenly
distributed across the western CONUS on a yearly basis.

For the relatively humid eastern states, we visually collected samples through interpretation of multi-temporal very high-
resolution images, street views, and time-series Landsat data on Google Earth and Google Earth Engine, based on the
appearance of irrigation infrastructure such as wells, pipes, center pivot towers, and circular field patterns. Detailed methods
of sample generation are described in Xie and Lark (2021).



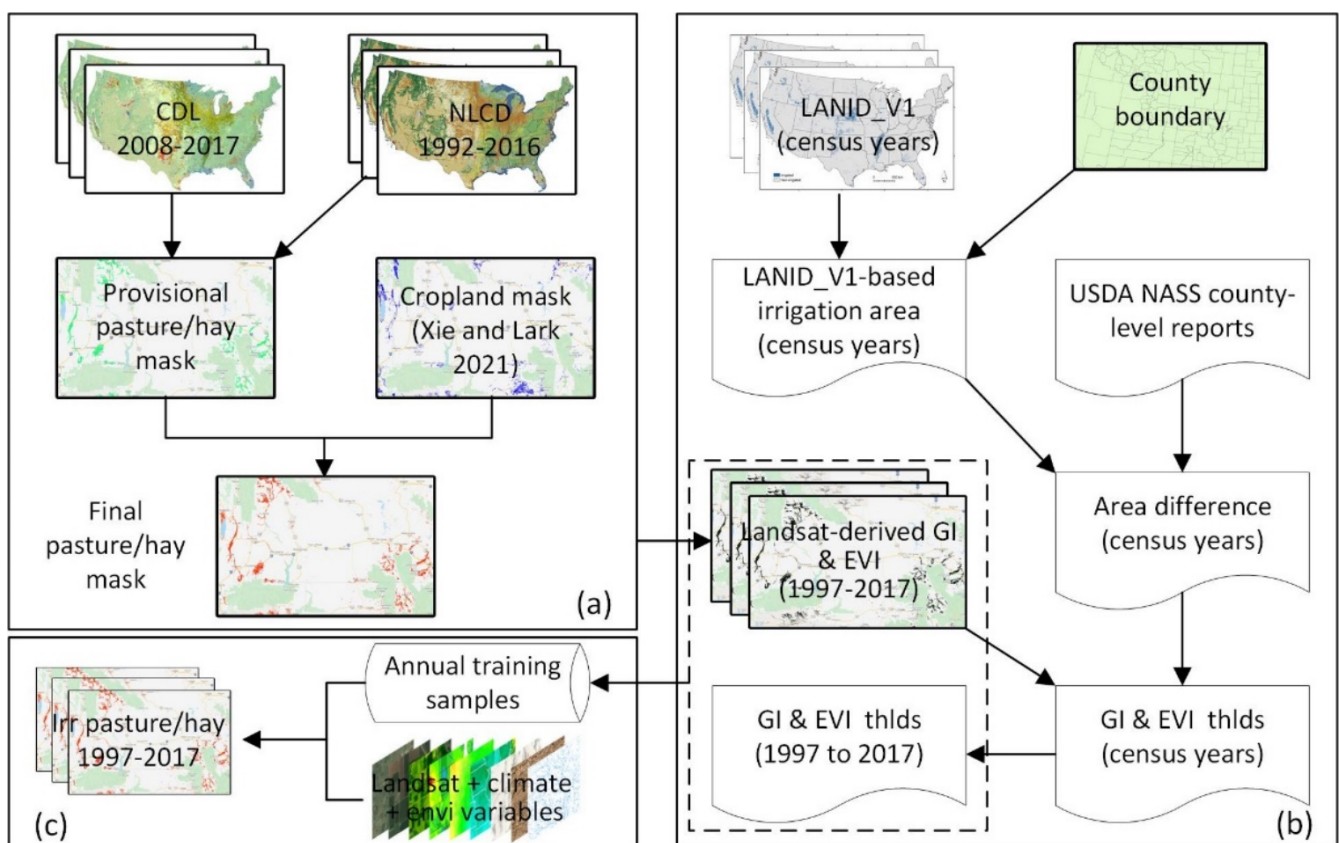

**Figure 1: Flowchart of mapping irrigated pasture and hay in western U.S. (a) generating the maximum extent of pasture/hay; (b) creating training samples, and (c) classification. Cropland mask refers to the maximum extent of non-pasture/hay cultivated land**
**created in Xie and Lark (2021).**

The predictors generally consist of two categories – satellite data and environmental variables (Xie and Lark, 2021). The primary satellite-derived variables were calculated from all available Landsat images within each year, including yearly maximum, median, and range composites of GI, EVI, and normalized difference water index (NDWI). Annual and late-season (May 1 to October 15) sum of MODIS-derived indices (i.e., EVI and land surface temperature) were also used as additional
variables. Environmental variables included annual and late-season sum of irrigation-relevant climate variables (i.e., precipitation, temperature, partial pressure of water vapor), elevation and slope, soil water content, and distance to major rivers (Deines et al., 2017; Deines et al., 2019; Xu et al., 2019; Xie et al., 2019). Altogether, there were 32 input features (25 for the years 1997-2000 when MODIS products were not available).

Classification was implemented on Google Earth Engine, a cloud-computing platform that enables rapid accessing and
processing of vast numbers of satellite images, climate data, and geophysical products (Gorelick et al. 2017). The classification was conducted annually per county using the widely used random forest classifier (Breiman 2001). The county-level classifications were mosaiced to create an initial time-series nationwide irrigation map, followed by logic and spatial filtering to remove possible false classification (see details in Xie and Lark (2021)).

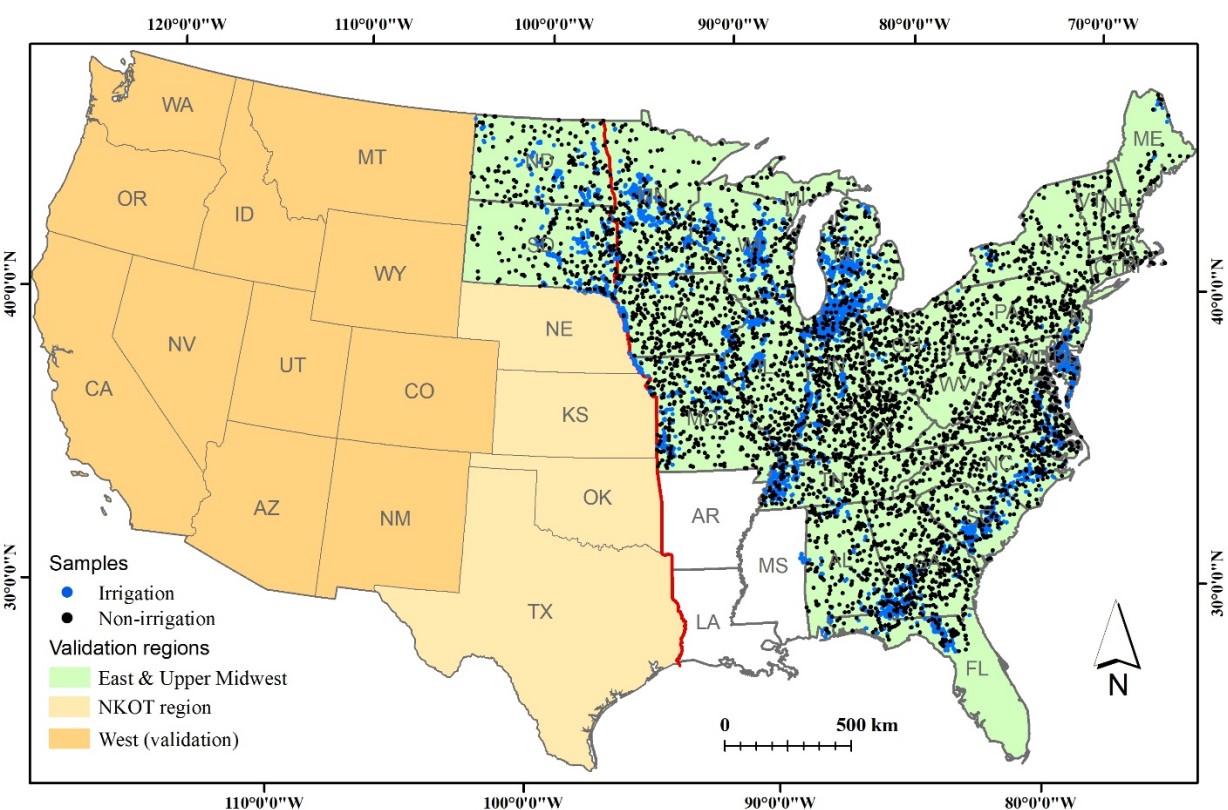

**Figure 2: Map evaluation and comparison design and the distribution of test sample locations across the eastern CONUS. The NKOT region refers to Nebraska, Kansas, Oklahoma, and Texas, which covers the majority of the High Plains Aquifer. The red solid line represents the West-East division for classification only.**

## 3 Map evaluation and comparison design

Comprehensive assessment of nationwide irrigation maps is not possible without adequate ground truth data, especially for the
eastern U.S. Therefore, accuracy of many published irrigation maps covering CONUS have been poorly evaluated. We compared our LANID maps to existing nationwide irrigation-specific maps, including two binary maps (i.e., MIrAD and GIAM) and two maps of irrigation fraction (i.e., MIF and GMIA areal percentage equipped for irrigation) (Table 1). Other global maps that include irrigation-related classes, such as Global Land Cover Map and GFSAD, are not shown because they are not irrigation-specific and substantially under-represent irrigation extent across CONUS. In addition to coarser resolution
nationwide maps, we also compared our maps with recently available 30-m resolution maps for the High Plain Aquifers and the eleven western states, i.e., AIM-HPA and IrrMapper, respectively.

Map evaluation and comparison were conducted by using test samples from two sources that cover the majority area of CONUS – a published reference dataset from Ketchum et al. (2020) and an additional independent dataset that we collected for this study. The test samples from Ketchum et al. (2020) were collected through visual interpretation of field parcels based on
irrigation clues from VHR images and crop greenness. The dataset has approximately 100,000 sample points, covering 11





western states (Fig. 2) for the whole study period of 1997-2017. Our independently collected validation samples (approximately 10,000 locations) covered the remaining areas except for Arkansas, Louisiana, and Mississippi. Lastly, we evaluated LANID's capability to detect irrigation change from pixel to state scales.

# 4 Results

## 4.1 Irrigation samples across the eastern U.S.

To validate our maps, we collected approximately 10,000 irrigation and rainfed samples for the east (~5,000 for each category) (Fig. 2). Each irrigation sample records a center pivot location and the presence of irrigation infrastructure during 1997-2017 (Fig. 3). In addition, we measured the radius of each center pivot irrigation system, i.e., the distance from its center to its field boundary. Note that the length of corner arms (designed for corner irrigation) was not measured (e.g., Field #1 in Fig. 3). Stable non-irrigation samples record the locations with clear evidence of no irrigation infrastructure during the entire mapping period. The average pivot radius for all samples collected in the Eastern CONUS was 330 meters, but distributed bimodally around approximately 200 and 400 meters, which correspond respectively to broader rectangular circumscribed crop fields of 40 and 160 acres.

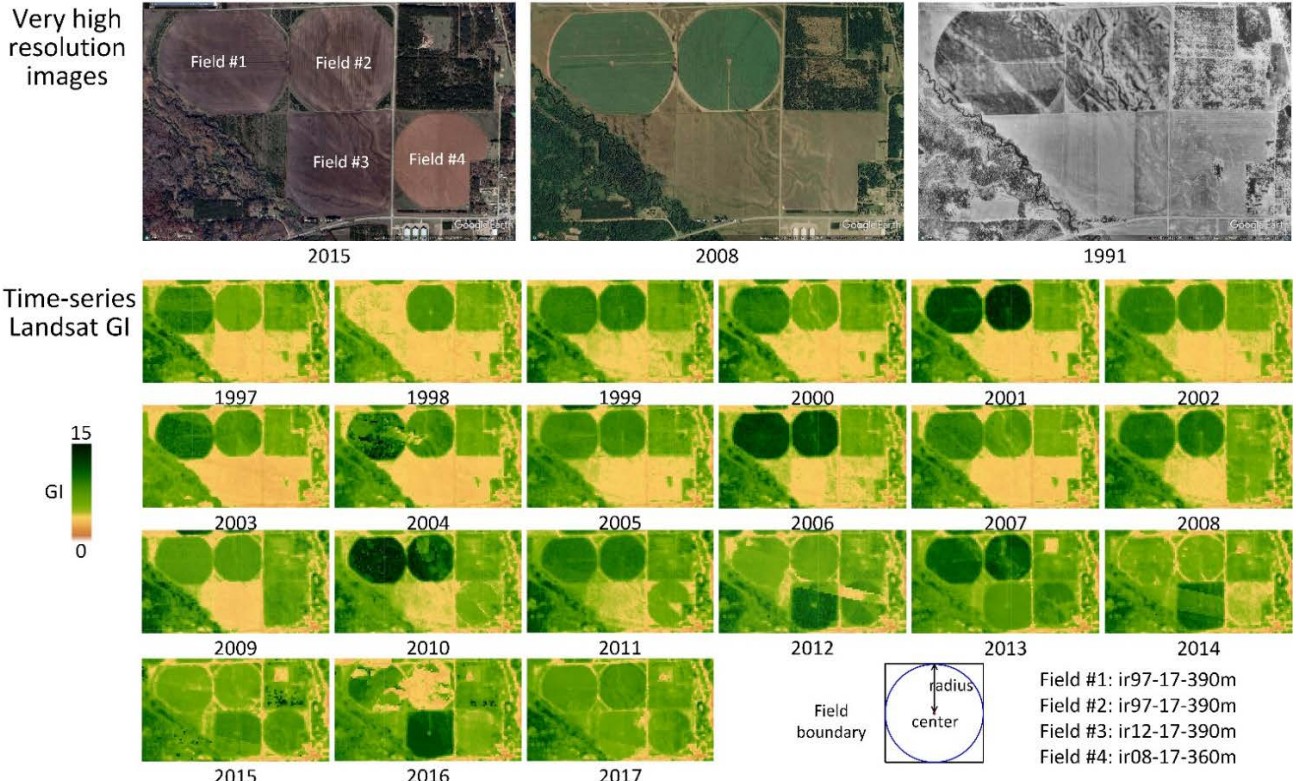

Figure 3: Demonstration of center-pivot irrigation field collection using time-series very high-resolution (© Google Earth Pro 2021) and Landsat images. GI: greenness index.

**4.2 Irrigation trends and changes**

Our LANID reveals a steady increase of irrigated area throughout CONUS, although there are some years with exceptional, lower values – for example, 2012 and 2002, years in which there were significant drought (Fig. 4) (Otkin et al., 2018). Overall,

irrigation area has increased by around 1.5 million hectares (Mha) during the period, from ~23 Mha before 2000 to ~24.5 Mha in 2016 with an average annual increase of ~80,000 ha. Consistent with earlier findings, the Central Valley of California, the High Plains portion of Texas, South-Central Florida, as well as select western states (e.g., Utah, Colorado, Idaho, and Wyoming) experienced substantial irrigation loss during the period (per-state plots in Fig. 4). In contrast, irrigation increased in states across the Midwest (including Nebraska, North Dakota, and South Dakota), the Mississippi Alluvial Plain, and the

East Coast. The largest gains occurred in Nebraska, Missouri, Michigan, Illinois, Arkansas, Mississippi, and Indiana, where irrigated area grew by over 100,000 ha per state.

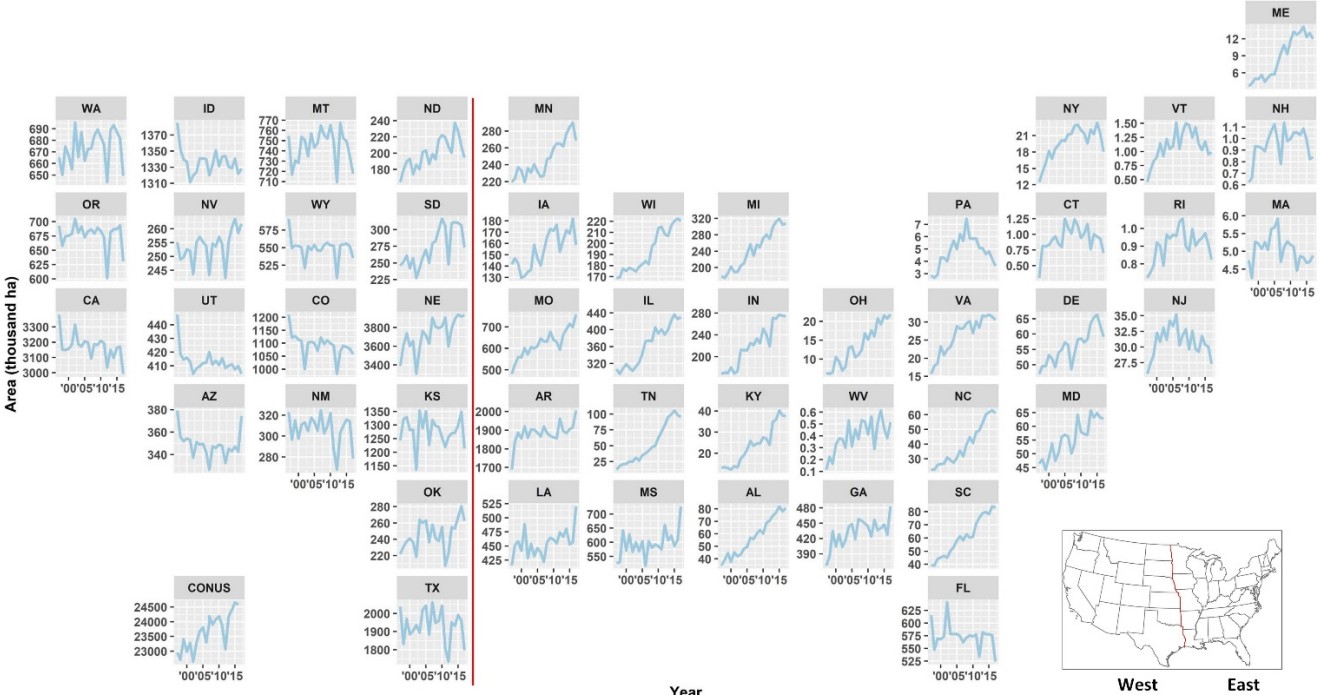

**Figure 4: LANID-derived annual irrigation area by state, 1997-2017. The red line shows the East-West division in this study based on a climatic transition near the 100th meridian. Annual irrigation area per state is provided in Table A1.**

Our LANID-derived irrigation changes agree well with USDA-NASS Census-reported values (with $R^2$ from 0.81 to 0.96), indicating that LANID and the USDA-NASS data are consistent in their detection of irrigation change at both county and state scales. Relative to the NASS data, however, our LANID maps predict slightly greater irrigated extent at the national level and slightly fewer net changes at both state and county levels, especially for the eastern CONUS (Fig. 5).

Aggregating the annual LANID maps to a finer but still intermediate 6-km resolution can reveal more localized trends than

state- or county-level data allow, while also accommodating for the field-level stochasticity and variations that often occur



Earth System
Science
Data

within a single farm or shared water source (Fig. 6). Such a resolution is particularly helpful for identifying small pockets of change with countervailing trends that would otherwise be masked or undetected. For example, we found outlier locations of irrigation loss in the Mississippi Alluvial Plain and of irrigation gain in the central and southern High Plains Aquifer.

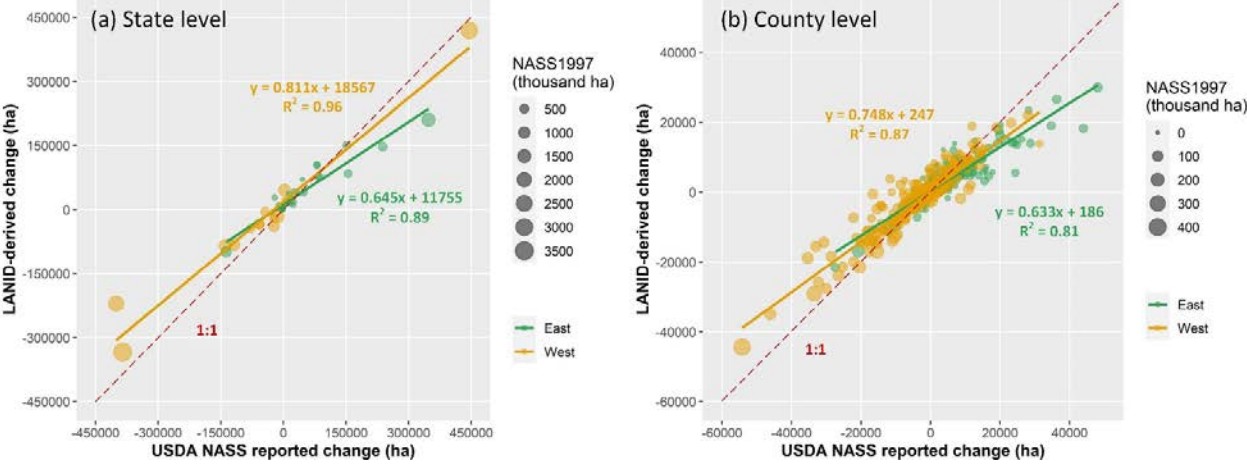

**180** **Figure 5: LANID-derived irrigation changes vs. USDA-NASS reported area at the state (a) and county (b) scale. Irrigation change was calculated as the difference between mean area of years 2012 and 2017 and that of 1997 and 2002 (i.e., mean(irArea$_{2012}$ + irArea$_{2017}$) – mean(irArea$_{1997}$ + irArea$_{2002}$) where irArea$_{yr}$ refers to irrigation area of year *yr*). The USDA-NASS reported values of 1997 is shown to represent irrigation area at the starting point of the study period.**

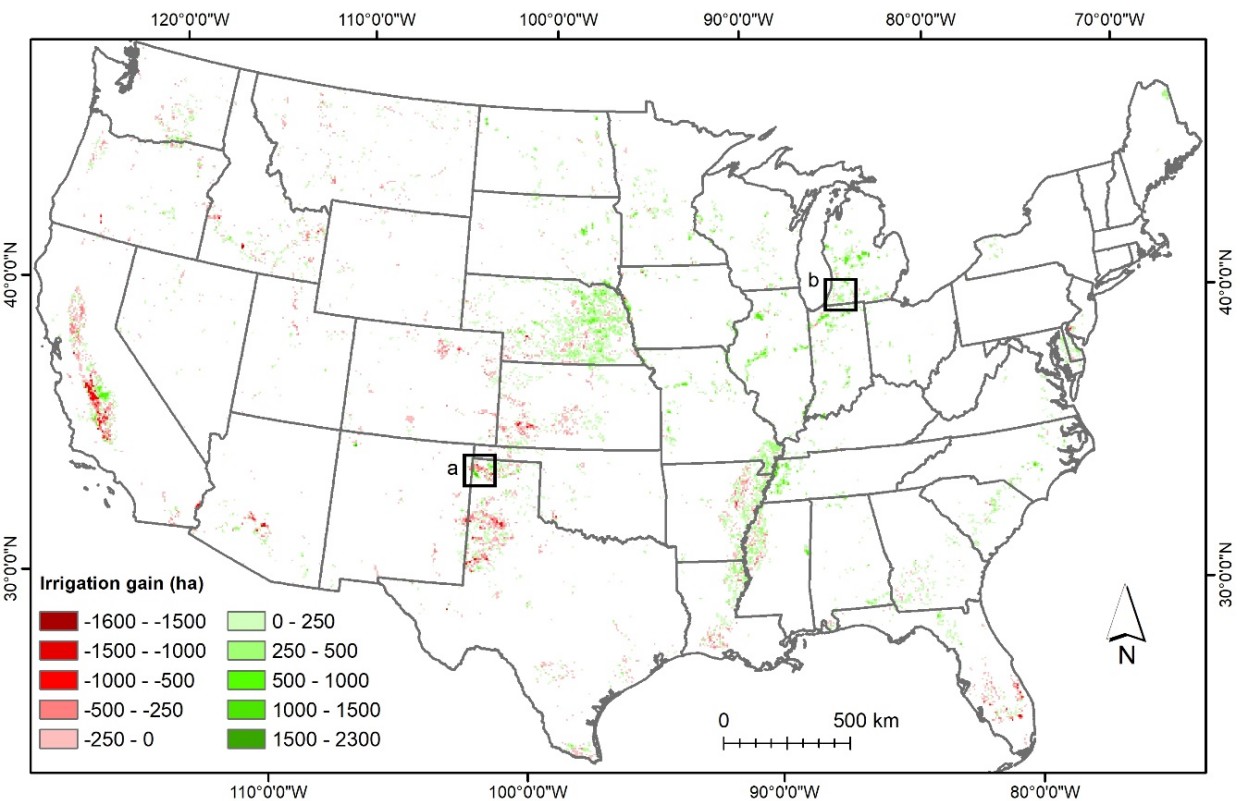

**Figure 6: LANID-derived irrigation gain from 1998-2007 to 2008-2017 at the 6km×6km scale. Per-grid value is calculated as the difference between mean irrigation area of 1998-2007 and that of 2008-2017 (i.e., meanIrArea2008-2017 - meanIrArea1998-2007, where irArea is LANID-aggregated irrigation area within a 6km×6km grid). Grids with absolute change < 2.5 % of are shown as background.**

Ultimately, when applied at the highest resolution, our LANID maps can be used to reliably characterize irrigation dynamics

at sub-field to field level with overall accuracy and Kappa index of 81 % and 0.62, respectively (Table 2). For instance, sub-field to field level expansions, losses, and interannual variations of irrigation that are detectable from LANID can be clearly observed in north Texas (Fig. 7a). Although such a level of change detection in more humid areas is not as effective as more arid states due to a weaker contrast between irrigated and rainfed fields, LANID still provides a reasonable and accurate characterization of irrigation change through time there as well, as shown in the example in Michigan (Fig. 7b).


**Figure 7: Demonstration of LANID-derived field-level irrigation frequency change for the northern Texas (a), and southwestern Michigan (b), respectively (highlighted in Fig. 6). Frequency change refers to the difference of number of years irrigated between 1998-2007 and that of between 2008-2017 (i.e., irFreq2008-2017 - irFreq1998-2007, where irFreq is the number of years irrigated).**

**Table 2. Accuracy of change detection using LANID maps. Change is defined as frequency difference between the two sub-periods**
**(i.e., 1998-2007 and 2008-2017) greater than 3 and the stable class refers to the value smaller or equaling to 3. Note non-agriculture is excluded from the stable class.**

| | | Reference | | |
|---|---|---|---|---|
| | | Stable | Change | User's accuracy |
| **Classified** | Stable | 187 | 63 | 75 % |
| | Change | 13 | 137 | 91 % |
| | Producer's accuracy | 94 % | 69 % | |
| | Overall accuracy: 81 %; Kappa: 0.62 | | | |

### 4.3 Irrigated pasture and hay

This study provides the first complete mapping and delineation of irrigated pasture and hay for the western U.S. (Fig. 8). In this region, forage and fodder crops provide valuable feed for livestock and irrigation is often necessary to cultivate certain
species or attain viable yields. This contrasts with pasture and hay in the eastern states, where annual precipitation and soil moisture is typically sufficient for robust production of grass-based forage and fodder. Areas of irrigated pasture and hay have a pattern of land use distinct from that of irrigated croplands, as well as unique implications for water use and the environment.

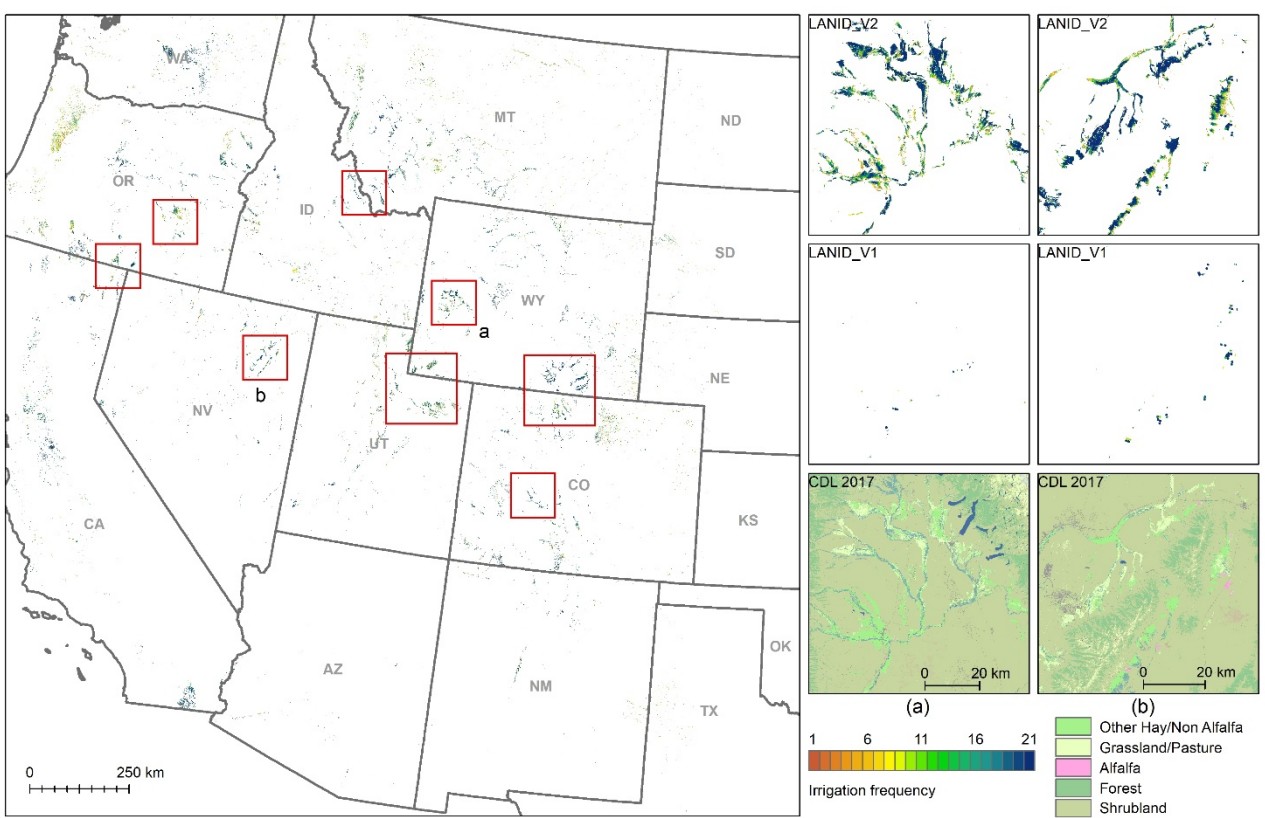



**Figure 8: Distribution of irrigated pasture and hay derived from LANID_V2 (presented in this study) in the western CONUS. The**
**overview display shows irrigation frequency (i.e., the number of years a pixel is irrigated during 1997-2017). The highlighted areas**
**under red rectangle represent areas of intensively irrigated pasture and hay that were not completely mapped in LANID_V1. (a)**
**and (b) are example local views for western Wyoming and northeastern Nevada, respectively.**

Compared to the first version of LANID, which did not explicitly include irrigated pasture/hay, we found an average of 0.34

Mha more irrigated land (i.e., more irrigated pasture and hay included in LANID_V2 compared to LANID_V1) for the years

2013 to 2017, and a similarly larger amount (0.36 Mha) since the start of the study period. This increase in irrigated extent is

lower than that of the USDA Census of Agriculture's estimate of 1 Mha of irrigated pasture – the only other spatial (but coarse)

estimate of such irrigated land use (Sanderson et al., 2012). The difference between our annual estimates and that of the

Census data likely reflects the fact that a large portion of irrigated pasture and hay (especially alfalfa) had already been mapped

in the first version of LANID. To confirm this, we further calculated a direct estimate of only irrigated pasture/hay as all

irrigated pixels classified as pasture or hay in the NLCD or CDL and estimated an average area of 1.39 Mha across the years

2008, 2011, 2013, and 2016. This estimate is 0.39 Mha higher than the 1 Mha reported by the Agricultural Census but includes

both pasture and hay, whereas the Census estimate is for pasture only.

## 4.4 Maximum extent, frequency, and formerly and intermittent irrigated land

Across all types of irrigation – including cultivated cropland and pasture and hay – a total of 38.5 Mha of land were irrigated

at least one time between 1997 and 2017, representing the maximum irrigated extent in the U.S. for our study period (Fig. 9a

and Table 3). Of these areas, just 24.2 Mha (62.8 %) were irrigated in 2017, and this annual utilization percentage ranged

from 58.8 to 64.0 % over the full study period. Across all pixels within the maximum irrigated extent, the mean annual

irrigated frequency was 12.9 out of 21 years (Fig. 9b). The distribution of irrigated frequency suggests many areas consist of

stable, persistent irrigation, but that there also exists a substantial amount of land with intermittent irrigation use. Those pixels

with the very lowest irrigation frequency likely reflect locations where irrigation ceased very early in the study period or was

first initiated very late in the study period, and/or areas of potential misclassification.

Looking at the subset of lands that are no longer irrigated, we found 4 Mha of formerly irrigated land (i.e., not irrigated anytime

in the most recent 3 years, 2015-2017, but that were irrigated at least 3 times prior) (Table 3). This formerly irrigated land is

primarily distributed across the western states (as showed in Fig. 6), and may reflect areas where insufficient water availability

has limited the ongoing use, or where salination of soils, socioeconomic drivers, or other superseding factors have resulted in

a cessation of irrigated agriculture. Of these formerly irrigated areas, 71.6 % remain in crop production under rainfed

conditions, primarily planted to corn (13.2 %), soybeans (12.3 %), and spring/winter wheat (12.2 %) as of 2017. The remaining

locations have either been abandoned from cultivated crop production altogether (26.3 %) or converted to urban use (2.1 %).

Those areas for which an irrigated crop is no longer viable may represent an opportunity for farmers to transition to grassland-

based agriculture (Deines et al., 2020), for example via the introduction of pasture for livestock grazing or the harvesting of

biomass for use as forage or cellulosic bioenergy feedstock (Robertson et al., 2017). As climate change and decreasing

freshwater availability continue to strain water resources, the total area of formerly irrigated lands is likely to increase, thereby



creating even further opportunity and greater need for alternative, drought resistant agricultural opportunities, such as those afforded by perennial feedstock production.

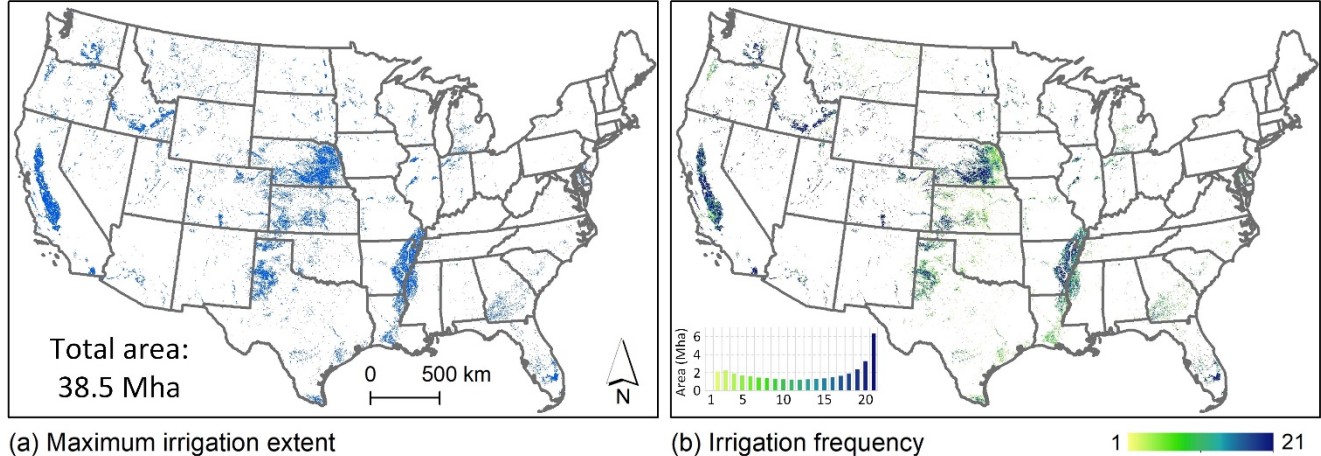

(a) Maximum irrigation extent

(b) Irrigation frequency


**Figure 9: The maximum irrigation extent (lands that have been irrigated at least once) and irrigation frequency (the number of irrigated years) across CONUS for the period 1997-2017. The inset in (b) shows the area of each frequency value.**

In addition to those locations where irrigation has ceased completely, we observed a substantial amount of land where irrigation remained active in the most recent years but where its use across time was discontinuous. For example, we found 25.5 Mha

of land across CONUS that had been irrigated spanning the whole study period (i.e., irrigated at least once for both 1997-1999 and 2015-2017), where over half of that subset (i.e., 13.5 Mha) could be best described as intermittently irrigated (frequency ≤ 18) (Table 3). As opposed to those locations with continuous annual irrigation use or where irrigation has ceased altogether, these intermittently irrigated lands appear to remain in irrigated agriculture today yet rely on such irrigation use just 67 % (median value) of the time across the 21-year study period. While further investigation is needed to better characterize these

areas of partial irrigation use over time, it may be possible that they represent locations where irrigation is only supplemental (e.g., used only in dry years or when needed), shared among a single water source but rotated among multiple nearby fields, or used only in years with sufficient water availability or water application rights and allocations. Similar to formerly irrigated lands, these locations of intermittent irrigation application may present areas of opportunity or economic need for alternative, rainfed agriculture in non-irrigated years. In such cases, drought tolerant annual crops like forage or energy sorghum could

potentially provide economic opportunities for producers and limited further strain on local hydrology.

**Table 3. Statistics of irrigation area (in million hectares) across CONUS for the period 1997-2017.**

|  | **Area** | **Definition** |
|---|---|---|
| Average annual area | 23.7 | Mean annual irrigation area |
| Maximum area | 38.5 | Irrigated at least once |
| Formerly irrigated | 4.0 | Not irrigated anytime in 2015-2017, but irrigated at least 3 times prior |



| Long-term irrigation | Intermittently irrigated | 13.5 | Irrigated at least once for both 1997-1999 and 2015-2017, and irrigation frequency ≤ 18 |
|---|---|---|---|
| | Continuously irrigated | 12.0 | Irrigated at least once for both 1997-1999 and 2015-2017, and irrigation frequency > 18 |

## 4.5 Comparisons with existing products

Figure 10 presents the nationwide view of a single year LANID as well as other irrigation-specific products. The 30-m LANID 2005 map was aggregated to 10-km resolution (Fig. 10b) for comparing with other coarser resolution maps. Across broad scales, all maps show similar irrigation hotspots of the High Plains Aquifer, the Central Valley Aquifer, the Mississippi Alluvial Plain, the Snake River Aquifer, and the East Coast. While it might be reasonable to conclude that all these coarse resolution maps can capture similar irrigation patterns at the national scale, regional views emphasize the details that are uniquely captured by LANID. For instance, LANID identifies fewer irrigated pixels at the eastern Columbia Plateau Aquifer than other maps, especially compared to MIF and GIAM (Fig. 11). In another example of the High Plains Aquifer, GIAM and MIF substantially overestimate irrigation extent in the western and central Kansas compared to both LANID and MIrAD (Fig. 12). Among all comparison products, MIrAD provides the most similarity of irrigation patterns as LANID in the arid to semi-arid West and Midwest.

In more humid areas like the upper Midwest, our LANID map captures patterns that are considerably misclassified by other maps (Fig. 13). For example, GIAM and MIF omit the majority of irrigated fields in the region; MIrAD shows a clear administrative boundary effect and near random distribution of irrigation within each county. At 10 km resolution, GMIA provides similar patterns as LANID but exaggerates the overall irrigation extent.

Locally, LANID shows a substantial improvement of spatial detail compared to other maps. For example, boundaries of center pivot and rectangular fields are clearly recognizable in LANID, while they are obscured even on the 250 m resolution MIrAD (insets (h) and (j) of Figs. 11 and 12). It is also evident that LANID shows comparable spatial details as other regional maps IrrMapper and AIM-HPA (inset (i) of Figs. 11 and 12) while still offering consistent and comprehensive coverage across the CONUS.



**Figure 10: Nationwide views of different irrigation mapping products. LANID 2005 is aggregated to 1 km (a) and 10 km (d) resolution for comparison purpose. The LANID-derived irrigation frequency refers to the number of years a pixel is classified as "irrigated".**


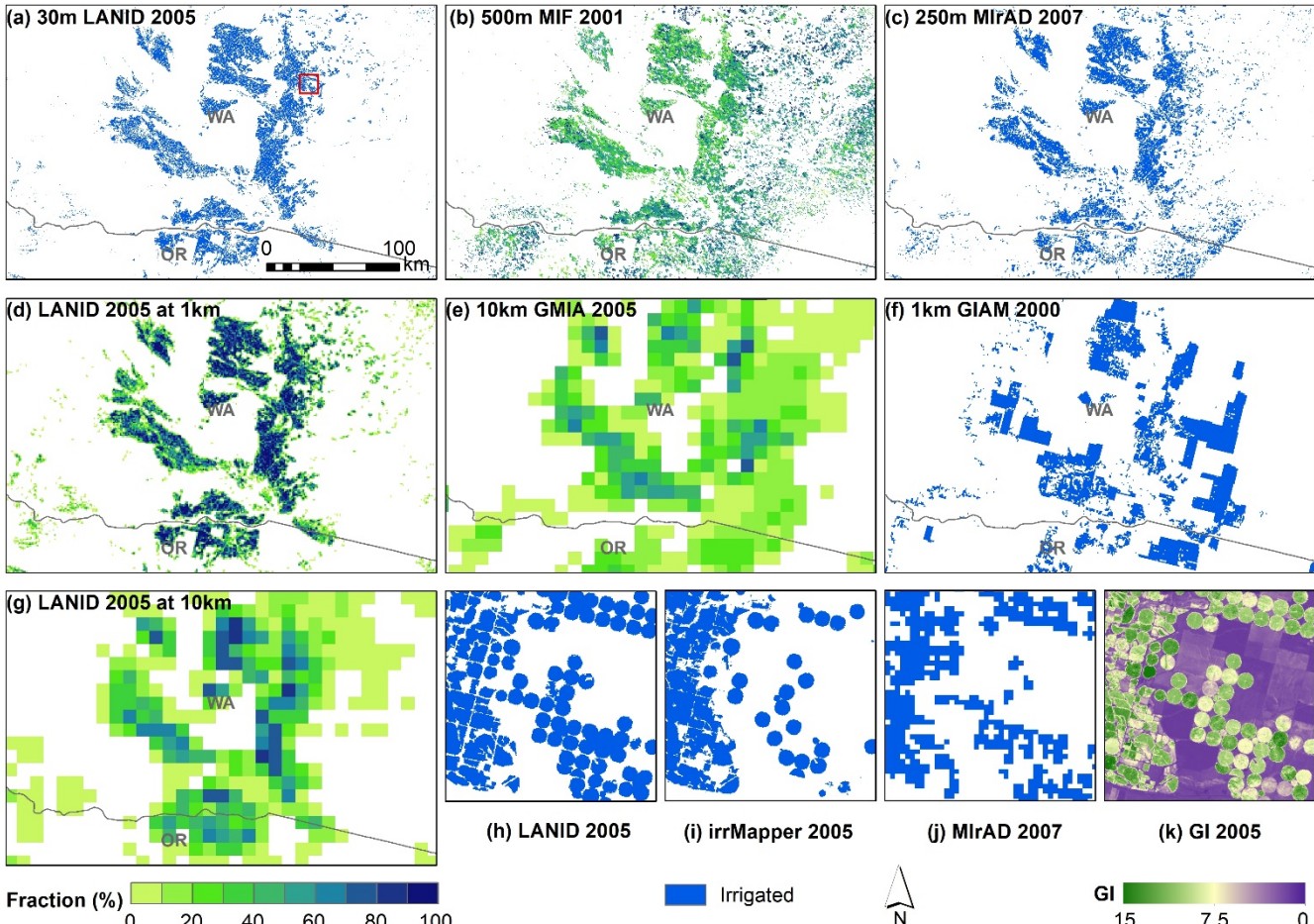

Figure 11: Product comparison at the Columbia Plateau Aquifer in northern Oregon and southern Washington. In addition to the original 30-m LANID (a), the map is aggregated to 1 km and 10 km resolution for displays (d) and (g). (h)-(i) show the location highlighted in (a) (red rectangle).



Figure 12: Product comparison at the High Plains Aquifer. In addition to the original 30-m LANID (a), the map is aggregated to 1 km and 10 km resolution for display (d) and (g). (h)-(i) show the location highlighted in (a) (red rectangle).

**Figure 13: Product comparison in central Minnesota. In addition to the original 30-m LANID (a), the map is aggregated to 1 km and 10 km resolution for display (d) and (g). (h)-(i) show the location highlighted in (a) (red rectangle).**

At the state level, our LANID estimates are consistent with USDA-NASS reported data (Fig. 14b), although the agreement is weaker than that of products like MIrAD and GMIA, which both rely directly and exclusively on census data as areal reference (not shown in the figure). In contrast, MIF underestimates irrigated area at the state level (Fig. 14c), whereas GIAM substantially overestimates irrigation extent especially for the states with reported area greater than one million hectares (Fig. 14d).

**Figure 14: Comparisons of irrigated area between products at the nation (a) and state (b-d) level. (a) LANID-derived nationwide irrigation trend (dashed pink line) and irrigated area of other products; (b) USDA-NASS reported vs. LANID-estimated irrigation area for five census years; (c) USDA-NASS reported (2002) vs. MODIS-estimated (2001) irrigated area (adapted from Ozdogan and**
**Gutman (2008)); (d) USDA-NASS reported (2002) vs. GIAM-estimated (2000) irrigated area. Note the GIAM-estimated nationwide irrigated area (39 million ha) is not shown in (a) due to its exceptionally high value. State-level comparisons between USDA-NASS and MIrAD-US and GMIA are not demonstrated because both products used census data as reference.**

The results of pixel-based assessment further reveal the advantages of LANID over other nationwide maps (Table 4). We find

that the overall accuracy is generally high for the NKOT region (i.e., Nebraska, Kansas, Oklahoma, and Texas) across all nationwide maps except for GIAM, with mean accuracy ranging from 78.9 % (MIF) to over 95 % (the LANID maps). Similarly, all maps show relatively high overall accuracy for the 11 western states, with values ranging from 82.6 % of MIF to 94.2 % of MIrAD. Despite these maps' reasonable accuracy in the west and even Midwest, they incorrectly assign a considerable number of rainfed fields as irrigated possibly due to coarse resolution and their difficulty separating them in some

areas such as the Columbia Plateau Aquifer (Fig. 11). For example, GIAM captures many low-density pixels in the west (Fig. 15c); MIF overestimates the locations with irrigation fraction between 0 and 60 % (Fig. 15b); MIrAD maps irrigated pixels with median fraction around 80 % (Fig. 15a).

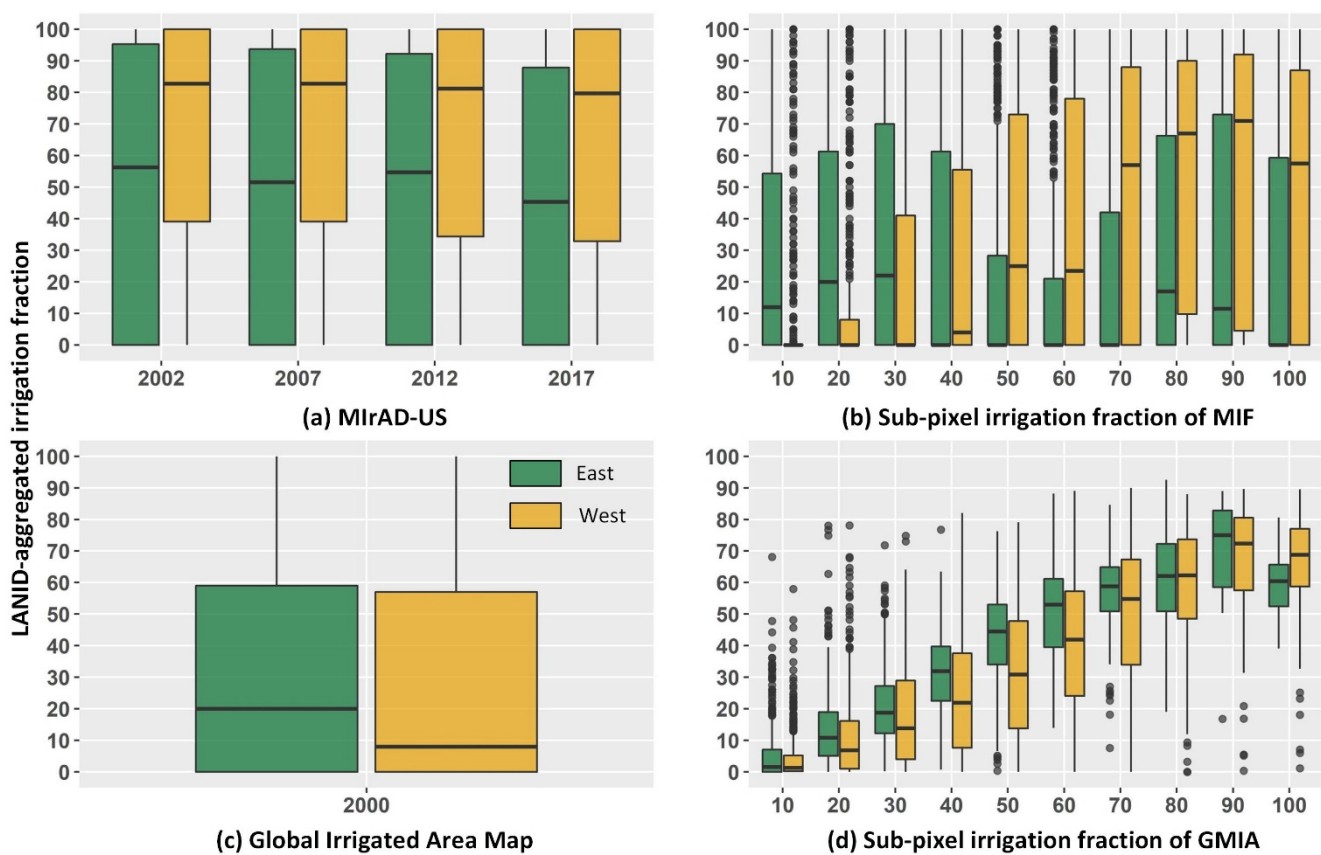

**Figure 15: Box plots showing irrigation fraction mapped in each product using LANID as reference. The western and eastern**
**CONUS (separated by red line in Fig. 2) are shown as brown and green, respectively. The 30-m LANID maps were aggregated as irrigation fraction to match the spatial resolution of each product (e.g., 250-m for MIrAD). For binary maps MIrAD and GIAM, five thousand irrigated samples were stratified for both west and east; fifty samples were selected for each irrigation fraction from 1 to 100 % (with increment of 1 %) in MIF and GMIA. The numbers on the horizontal axes of (b) and (d) refer to the maximum value of each bin.**



In the eastern U.S., our LANID maps stand out with overall accuracy of 94.4 % – on par with their performance in the western U.S. – whereas other maps show accuracy below 60 %. The extremely low accuracy of MIrAD, MIF, and GIAM in the east is attributable to their missing of most irrigated cropland as well as frequent false identification of rainfed cropland as irrigated (see Fig. 13 as an example), as characterized by omission error rates over 80 % and commission error rates over 45 % for the "irrigated" and "non-irrigated" class, respectively. As a result, MIrAD maps irrigated pixels in the east that have a median

irrigated fraction of about 50 % according to LANID (Fig. 15a); GIAM misclassifies a substantial number of low-density pixels (Fig. 15c); MIF substantially overestimates the locations with irrigation fraction beyond 30 % (Fig. 15b).

We also compared our maps to AIM-HPA (i.e., Annual Irrigation Maps – High Plains Aquifer) (Deines et al., 2019), a dataset with the same spatial and temporal resolution as LANID but covering only the High Plains Aquifer. In this region, LANID performs comparably to the HPA-specific dataset, with overall accuracy of 95.9 % vs. 93.2 %, respectively, and Kappa values

of 0.89 vs. 0.82.

For a broader 11 western state region, our LANID maps show 92.8 % congruence (Kappa of 0.84) with the reference data from IrrMapper (Ketchum et al., 2020) compared to a 99.1 % (Kappa of 0.98) congruence of the IrrMapper product with its reference data. Such results follow in part from the methods of reference data utilization, as IrrMapper used 60 percent of the validation data used in our comparison for its classifier training.  Further differences between LANID and IrrMapper may stem from

differences in sampled data and irrigated class definition.  For example, the IrrMapper point-based irrigation samples were stratified from verified fields that were digitized in years different from the time of irrigation verification, such that they likely capture permanently irrigated croplands well but may potentially include fields that are partially irrigated or fallowed in any given year. In addition, IrrMapper's reference irrigation samples appear to include both irrigated croplands and other grass-like lands, such as irrigated turfgrass and groundwater- or fluvial-subsidized grasslands and wetlands.  This broader and more

variable pool of reference data may thus help explain additional observed differences, such as occasionally less distinct field boundaries in IrrMapper as compared with LANID and GI (e.g., lefthand portions of Fig. 11h-k) as well as the slightly higher apparent accuracy of MIrAD (which relies only on vegetation greenness) compared to LANID in the west when assessed against the IrrMapper reference data (Table 4).  Thus, while overall performances of LANID and other datasets are similar in overlapping regions like the HPA and the western states, differences in each product's intent and class specificity will likely

dictate preferences for specific user applications.

**Table 4. Confusion table of pixel-wise accuracy assessment. The overall accuracy, omission error (1 – Producer's accuracy), and commission error (1 – User's accuracy) are in percent. Accuracy values are averaged if multiple-year assessment was conducted. Parenthetical numbers represent the standard deviation.**

| Maps | Region | Year | Kappa | Overall accuracy | Omission error | | Commission error | | Sample size | Irrigation sample |
|---|---|---|---|---|---|---|---|---|---|---|
| | | | | | Irrigated | Non-irr | Irrigated | Non-irr | | |
| LANID | West[a] | 1997-2017 | 0.84 (0.07) | 92.8 (3.5) | 11.4 (6.9) | 3.6 (1.0) | 6.1 (4.9) | 10.7 (8.9) | 4433 | 2284 |
| | NKOT | 1997-2017 | 0.93 (0.02) | 96.6 (0.8) | 5.9 (1.4) | 1.0 (0.2) | 1.0 (0.2) | 5.6 (1.3) | 9994 | 5002 |
| | East | 1997-2017 | 0.89 (0.01) | 94.4 (0.6) | 10.7 (1.2) | 0.5 (0.1) | 0.6 (0.1) | 9.7 (1.0) | 10000 | 5000 |
| | HPA[b] | 1997-2017 | 0.89 (0.03) | 95.9 (1.1) | 4.7 (1.4) | 2.3 (0.6) | 0.7 (0.2) | 13.0 (3.3) | 5890 | 4479 |



|  | | | | | | | | | | |
|---|---|---|---|---|---|---|---|---|---|---|
| | West[a] | 2002, 2007 | 0.84 (0.02) | 94.2 (0.8) | 10.3 (2.4) | 4.3 (0.9) | 11.2 (5.4) | 4.3 (2.1) | 3102 | 987 |
| MIrAD | NKOT | 2002, 2007, 2012, 2017 | 0.76 (0.05) | 87.8 (2.5) | 18.0 (4.4) | 6.3 (0.7) | 7.1 (1.0) | 16.2 (3.4) | 9967 | 5014 |
| | East | 2002, 2007, 2012, 2017 | 0.16 (0.01) | 58.0 (0.7) | 82.3 (1.1) | 1.7 (0.6) | 8.7 (2.9) | 45.6 (0.4) | 10000 | 5000 |
| | West[a] | 2001 | 0.49 | 82.6 | 47.8 | 7.4 | 29.9 | 14.6 | 3002 | 747 |
| MIF[c] | NKOT | 2001 | 0.58 | 78.9 | 27.2 | 14.9 | 17.0 | 24.3 | 9985 | 5001 |
| | East | 2001 | 0.12 | 55.9 | 83.6 | 4.6 | 21.9 | 46.7 | 10000 | 5000 |
| | West[a] | 2000 | 0.72 | 87.6 | 23.5 | 6.2 | 12.6 | 12.3 | 3436 | 1234 |
| GIAM | NKOT | 2000 | 0.25 | 62.6 | 57.2 | 17.5 | 29.0 | 41.0 | 10040 | 5023 |
| | East[a] | 2000 | 0.04 | 52.2 | 93.5 | 2.0 | 23.6 | 48.8 | 10000 | 5000 |
| AIM-HPA | HPA[b] | 1997-2017 | 0.82 (0.04) | 93.2 (1.8) | 6.9 (2.4) | 6.4 (3.4) | 2.1 (1.1) | 18.6 (4.8) | 5890 | 4479 |
| | HPA[d] | 1997-2017 | - | 92.7 (1.5) | 14.0 (4.5) | 3.1 (1.7) | 8.5 (2.1) | 8.5 (2.1) | 1316 | 519 |
| IrrMapper | West[a] | 1997-2017 | 0.98 (0.01) | 99.1 (0.3) | 0.3 (0.2) | 1.4 (0.3) | 2.4 (1.9) | 0.3 (0.2) | 4433 | 2284 |
| LANID2012 | NKOT | 2012 | 0.84 | 92.0 | 10.1 | 5.8 | 6.0 | 9.8 | 9938 | 5002 |
| | East | 2012 | 0.49 | 74.4 | 49.4 | 1.9 | 3.7 | 33.5 | 10000 | 5000 |

[a] Validation samples from Ketchum et al. (2020). Test samples for the years 1999, 2004, 2005, 2012, 2015, and 2017 were not used because
of limited irrigated samples. [b] Validation samples from this study. [c] Irrigated pixels were set as fraction greater than 20 %. [d] Accuracy
assessment reported by Deines et al. (2019). NKOT: Nebraska, Kansas, Oklahoma, and Texas.

## 5 Discussion

### 5.1 Uncertainty, limitations, and future improvements

Both qualitative and quantitative assessments show extensive improvements of LANID compared to other currently available
nationwide maps in terms of spatial detail and temporal frequency. Despite the advances, caution is still needed especially
when applying the dataset at the scale of individual fields in the eastern U.S. For example, mapping accuracy in the MAP
region is uncertain due to the absence of reference data and the difficulty of collecting aerial ground truth in the area. In
addition, map accuracy in the humid East is slightly lower than in the arid and semi-arid West. The quality of maps might also
vary over time due to availability of clear Landsat observations. For instance, fewer Landsat images in 2012 constrained map
quality and scan-off effects of the ETM+ sensor might remain in some areas.

We took several post-classification steps to improve mapping accuracy, which also introduces limitations to LANID. First,
our minimum mapping unit of 5 acres (i.e., 23 Landsat pixels) improved mapping confidence but also excluded smaller
irrigated fields, such as fragmented irrigated vegetable fields often found in suburban and peri-urban areas. Second, the
assumption that fields equipped with irrigation systems tend to be cropped and irrigated frequently could have incorrectly
masked out some irrigated fields historically under long-term and frequent fallow (e.g., irrigated – long-term fallow – irrigated).
Lastly, our current version of LANID covers only the period of 1997 to 2017, which might be problematic for users who want



maps outside the study period. However, we hope to regularly update the existing dataset in the future to include the most recent years of available imagery, and, if able, extend the time series back in time through the duration of the Landsat record. Given these uncertainties and limitations, future generations of LANID could benefit from the following improvements. First,

we anticipate using our temporally extendable methodology to routinely update LANID, such that coverage could extend prior to 1997 and up to the most recent year. Efforts could also be made to enhance spatial detail (e.g., 10 m resolution) and mapping accuracy, particularly in the humid eastern U.S. where contrasts between irrigated and rainfed crop are obscure. This is practical for recent years when both the revisit frequency and spatial resolution of satellite observations are greatly improved. Lastly, implementation of an irrigation-specific change detection algorithm could help improve the identification and

consistency of monitoring variations in irrigation over time.

## 5.2. Potential applications

Our annual 30-m resolution nationwide LANID maps may be valuable to local, state, and regional water governance bodies, agribusinesses, and the research community for a variety of applications including water use estimation, risk assessment, use as model input, and more.

Our LANID maps could benefit water and agricultural managers by providing insights into irrigation changes (e.g., expansion and abandonment) at geographic and temporal scales relevant to decision-making. Our field-scale, wall-to-wall data will enable local and regional water management organizations, which may not otherwise have sufficient data or resources, to make better decisions that influence regional water availability. For example, state-level water managers and engineers who need to plan how much water to allocate for agriculture could utilize our irrigation distribution and change information to estimate demand.

Policy makers may also use LANID to navigate future decision making and to evaluate federal agricultural, bioenergy, and conservation policies (Mccarthy et al., 2020; Lark, 2020).

Our dataset may also be useful for agribusinesses and entities across agricultural supply chains. For example, our maps could be used by companies that seek to reduce risk from water scarcity within their supply chains or lower the water footprint of their sourced products (Brauman et al., 2020). Additional applications may include business decision-making and financial

investment (Turral et al., 2010), precise field-level water use estimation and solutions (Sadler et al. 2005), and crop yield prediction and its water resilience (Troy et al., 2015).

A key informant and collaborator in the development of our LANID maps has been the USGS, and the produced outputs may help support several ongoing USGS efforts, such as the National Water Census's efforts to provide water budgets at the watershed level (USGS, 2020c), the National Water-Use Information Program (NWUIP) dissemination of water use data

(USGS, 2020a), and the Water Availability and Use Science Program (WAUSP) assessments of regional groundwater availability (USGS, 2020b). The research community within USGS also has high priority goals to improve quantification of crop consumptive water use and project future water use. Our improved estimates of irrigation location, extent, and dynamics could help refine evapotranspiration estimates of irrigated croplands, thereby improving estimates of agricultural water use



from field to aquifer scale and further supporting the ongoing expansion of detailed water use estimates across the continental

U.S. (Senay et al., 2016; Senay et al., 2017).

We also hope that our dataset will serve several needs in the broader research community, especially for those who study hydrology, agriculture, and the environment from local to nationwide scales. Users of previous coarser resolution irrigation datasets will benefit from the improvements in spatial detail, product frequency, and map accuracy. Existing nationwide irrigation datasets like MIrAD have been accessed by hundreds of users in academia and government via the USGS EROS

website (Brown and Pervez, 2014). These data have been incorporated into studies to map cropland and its change, evaluate quality trends in ground and surface waters, model evapotranspiration and energy-water exchange at the surface boundary layer, and reveal locations at risk of unsustainable irrigation (Brown and Pervez, 2014; Pryor et al., 2016; Seyoum and Milewski, 2016; Jin et al., 2011; Zaussinger et al., 2019). Our 30-m data products will enhance similar types of applications and enable many others through the improved spatial and temporal resolution. To this extent, several organizations have begun

using our previously published LANID 2012 for further research and development activities, despite there being only 1 of the presently described 21 annual years of data available; such applications should be further enabled by the current full suite of products and time periods.

Lastly, our collected samples could help generate new threads of irrigation maps for the eastern U.S. Because insufficient ground reference data has long been a bottleneck to producing accurate classifiers for irrigation mapping, our verified locations

could facilitate the development and evaluation of new models for irrigation detection, especially when other constraints are becoming relieved due to increasingly available high- to moderate-resolution remote sensing images, development of machine learning algorithms, and open access of cloud computing platforms.

## 6. Data availability

Our annual LANID maps, their byproducts (i.e., maximum irrigation extent, irrigation frequency, and per-pixel irrigation

trends), ~10,000 manually collected ground reference data, and metadata can be accessed via https://doi.org/10.5281/zenodo.5003976 (Xie et al., 2021). All maps are projected to the "Albers Conical Equal Area" projection at 30-m resolution except for the map of irrigation trends of 6-km.

## 7 Conclusions

This paper presents the only annual, nationwide fine-resolution maps of irrigation extent for the U.S., which are available for

each year 1997-2017 and offer several improvements over other products. The increased resolution of the described LANID dataset sets a new standard in spatial detail at the CONUS extent, while the increased mapping frequency and multidecadal coverage enable characterization of irrigation dynamics. Our accuracy assessment shows that the LANID maps provide the




most realistic depiction of irrigation extent across the country, with performance that matches or exceeds existing regional datasets.

Moving forward, the LANID maps provide a foundation for refined representations of irrigation distribution and dynamics across the U.S. It is clear from recent research efforts that high quality, frequently updated data on fine-scale irrigation extent is immensely valuable for both the research and application user communities. With these needs in mind, our future intents and interests surrounding LANID may focus on: (1) routinely updating annual maps after 2017; (2) providing finer resolution maps of irrigation extent (e.g., 10m) by fusing multi-source imagery; and (3) improving mapping accuracy in the eastern

CONUS.

## Appendices

**Table A1. The LANID-derived state-level irrigated area (in hectares) of each year between 1997 and 2017 (1997-2008).**

| States | 1997 | 1998 | 1999 | 2000 | 2001 | 2002 | 2003 | 2004 | 2005 | 2006 | 2007 |
|---|---|---|---|---|---|---|---|---|---|---|---|
| Alabama | 34535 | 39125 | 44618 | 37601 | 45173 | 42095 | 43512 | 48210 | 49533 | 56837 | 55946 |
| Arizona | 379287 | 355684 | 352011 | 354327 | 353060 | 337113 | 351052 | 348990 | 349530 | 342038 | 326403 |
| Arkansas | 1687799 | 1839858 | 1886084 | 1855790 | 1920171 | 1859224 | 1903682 | 1902655 | 1886089 | 1866404 | 1919269 |
| California | 3380488 | 3151686 | 3149145 | 3156261 | 3184521 | 3314552 | 3187264 | 3172349 | 3206905 | 3197729 | 3094285 |
| Colorado | 1210321 | 1121823 | 1127185 | 1114290 | 1108564 | 1000578 | 1102934 | 1105161 | 1100720 | 1070917 | 1121951 |
| Connecticut | 302 | 807 | 807 | 829 | 910 | 970 | 851 | 801 | 1251 | 1108 | 1013 |
| Delaware | 47054 | 49638 | 49591 | 53128 | 52042 | 49221 | 53965 | 54924 | 57305 | 56429 | 48586 |
| Florida | 615570 | 547328 | 569116 | 568295 | 574083 | 640918 | 578654 | 578092 | 578447 | 574716 | 561719 |
| Georgia | 366690 | 384742 | 434499 | 404428 | 428114 | 403744 | 416991 | 442892 | 448528 | 418726 | 457852 |
| Idaho | 1385130 | 1352224 | 1339614 | 1336683 | 1311422 | 1319562 | 1323663 | 1340630 | 1340766 | 1339902 | 1319951 |
| Illinois | 306516 | 295264 | 308291 | 318324 | 309793 | 302484 | 312484 | 322255 | 343074 | 372371 | 374335 |
| Indiana | 168572 | 170884 | 170226 | 180406 | 169450 | 173272 | 212469 | 211910 | 211605 | 224793 | 221168 |
| Iowa | 141592 | 146916 | 142392 | 129494 | 131310 | 134782 | 136684 | 158873 | 146813 | 140562 | 153320 |
| Kansas | 1243244 | 1321112 | 1329850 | 1282021 | 1283028 | 1135702 | 1353488 | 1288916 | 1350727 | 1227553 | 1318920 |
| Kentucky | 13104 | 13479 | 12991 | 12209 | 13955 | 13346 | 17538 | 19164 | 21688 | 25816 | 23551 |
| Louisiana | 415211 | 451161 | 458231 | 442128 | 488285 | 428395 | 453754 | 432163 | 445974 | 438908 | 421933 |
| Maine | 3644 | 4142 | 4983 | 4910 | 5629 | 4403 | 5167 | 5738 | 5731 | 7701 | 9487 |
| Maryland | 46450 | 47965 | 44192 | 48584 | 53744 | 47490 | 50384 | 56148 | 56940 | 56354 | 50152 |
| Massachusetts | 4756 | 4250 | 5274 | 5216 | 5069 | 5258 | 5044 | 5629 | 5670 | 5920 | 4709 |
| Michigan | 179126 | 174745 | 186417 | 202131 | 189431 | 189615 | 204443 | 210328 | 233991 | 256992 | 229009 |
| Minnesota | 219513 | 223631 | 236929 | 233381 | 219645 | 236085 | 230768 | 240468 | 232043 | 225834 | 227401 |
| Mississippi | 526481 | 529629 | 641770 | 570773 | 627065 | 567028 | 599802 | 568687 | 603644 | 517243 | 604432 |
| Missouri | 480854 | 523644 | 559171 | 558573 | 600994 | 571590 | 605685 | 600251 | 610383 | 646444 | 632748 |
| Montana | 754784 | 716614 | 730501 | 728447 | 753781 | 750609 | 734975 | 756981 | 743386 | 747750 | 764726 |



| | | | | | | | | | | | |
|---|---|---|---|---|---|---|---|---|---|---|
| Nebraska | 3388881 | 3607761 | 3737425 | 3602001 | 3655262 | 3303885 | 3577851 | 3763770 | 3713858 | 3628089 | 3902377 |
| Nevada | 255173 | 248903 | 249718 | 252576 | 252090 | 243633 | 255082 | 257084 | 255298 | 253798 | 243299 |
| New Hampshire | 625 | 661 | 934 | 930 | 918 | 891 | 991 | 1065 | 1123 | 976 | 782 |
| New Jersey | 25712 | 27451 | 28717 | 32253 | 30855 | 33095 | 31175 | 34197 | 33369 | 35184 | 30257 |
| New Mexico | 323098 | 296609 | 314995 | 297562 | 310916 | 312938 | 304907 | 317588 | 313720 | 305273 | 324526 |
| New York | 12503 | 14324 | 15970 | 18173 | 16761 | 18620 | 19341 | 20163 | 20029 | 21209 | 21336 |
| North Carolina | 22569 | 22641 | 25847 | 26482 | 26672 | 30930 | 29181 | 27337 | 30732 | 35506 | 31644 |
| North Dakota | 164616 | 179741 | 190039 | 192953 | 174264 | 185754 | 180438 | 201479 | 203502 | 186949 | 199093 |
| Ohio | 6345 | 6091 | 6492 | 10584 | 9362 | 6949 | 7985 | 12980 | 13365 | 10487 | 11161 |
| Oklahoma | 221859 | 230949 | 237608 | 240967 | 236309 | 218499 | 263921 | 260352 | 263092 | 236626 | 257870 |
| Oregon | 693065 | 657674 | 674871 | 675721 | 678911 | 705284 | 681086 | 691636 | 672903 | 683382 | 686578 |
| Pennsylvania | 2877 | 2648 | 2918 | 4303 | 4295 | 4006 | 4970 | 5664 | 5006 | 6213 | 5665 |
| Rhode Island | 721 | 755 | 790 | 921 | 903 | 791 | 965 | 944 | 965 | 958 | 1039 |
| South Carolina | 39571 | 38942 | 44011 | 45249 | 46390 | 45040 | 50078 | 53012 | 57992 | 61285 | 58224 |
| South Dakota | 245846 | 251604 | 260802 | 241994 | 256887 | 227293 | 242316 | 257936 | 269721 | 250017 | 279162 |
| Tennessee | 12096 | 18410 | 20843 | 21540 | 24923 | 25049 | 30954 | 26660 | 34988 | 37998 | 42068 |
| Texas | 2037060 | 1832083 | 1970036 | 1886396 | 1902613 | 1935970 | 1894429 | 2020246 | 2042627 | 1883622 | 2061213 |
| Utah | 447520 | 418494 | 414730 | 415852 | 412975 | 404106 | 408105 | 409668 | 411897 | 412424 | 420019 |
| Vermont | 459 | 675 | 827 | 905 | 1143 | 918 | 1219 | 1048 | 1090 | 1508 | 1047 |
| Virginia | 15675 | 17807 | 18575 | 23268 | 20898 | 22374 | 23122 | 25289 | 28893 | 28187 | 28249 |
| Washington | 665353 | 650243 | 674586 | 666842 | 654969 | 695216 | 665695 | 687093 | 662436 | 672477 | 672925 |
| West Virginia | 115 | 224 | 165 | 328 | 378 | 373 | 300 | 531 | 312 | 455 | 373 |
| Wisconsin | 168788 | 169200 | 177363 | 175174 | 177969 | 176654 | 174700 | 179013 | 181334 | 183905 | 181335 |
| Wyoming | 591280 | 549623 | 552916 | 552251 | 550526 | 520762 | 552083 | 546981 | 553945 | 546109 | 545747 |
| CONUS | 22952830 | 22709864 | 23405066 | 22983454 | 23276428 | 22647066 | 23286147 | 23673951 | 23802940 | 23301684 | 23948855 |

**Table A1. Continued (2009-2017).**

| States | 2008 | 2009 | 2010 | 2011 | 2012 | 2013 | 2014 | 2015 | 2016 | 2017 |
|---|---|---|---|---|---|---|---|---|---|---|
| Alabama | 60139 | 63088 | 60198 | 68873 | 70572 | 74533 | 76987 | 81990 | 78022 | 80805 |
| Arizona | 347354 | 345935 | 348800 | 347701 | 332402 | 345229 | 342976 | 347084 | 342758 | 374399 |
| Arkansas | 1884322 | 1869713 | 1862017 | 1856051 | 1962000 | 1895565 | 1884232 | 1904854 | 1915934 | 2005406 |
| California | 3188969 | 3183257 | 3210136 | 3193344 | 3034074 | 3150194 | 3098087 | 3165015 | 3171492 | 2993121 |
| Colorado | 1097294 | 1111432 | 1099495 | 1090921 | 982058 | 1069911 | 1090256 | 1085316 | 1079062 | 1058369 |
| Connecticut | 1240 | 1146 | 946 | 1011 | 1167 | 766 | 1000 | 964 | 930 | 704 |
| Delaware | 53855 | 58197 | 58583 | 57379 | 58263 | 63623 | 65377 | 66345 | 62686 | 59182 |
| Florida | 571469 | 575663 | 574039 | 577377 | 533861 | 581562 | 577921 | 578069 | 575584 | 524110 |
| Georgia | 454795 | 449800 | 443450 | 423956 | 456294 | 436233 | 440079 | 447356 | 427585 | 482965 |
| Idaho | 1332914 | 1350822 | 1331055 | 1342894 | 1343860 | 1330616 | 1328663 | 1340649 | 1321842 | 1328081 |



| | | | | | | | | | |
|---|---|---|---|---|---|---|---|---|---|
| Illinois | 373296 | 405632 | 391193 | 401018 | 388723 | 400631 | 422017 | 435894 | 425569 | 429765 |
| Indiana | 232998 | 224904 | 251318 | 239992 | 219071 | 271049 | 269836 | 277084 | 274989 | 274193 |
| Iowa | 165879 | 172773 | 171304 | 176592 | 152614 | 162926 | 171756 | 168785 | 181923 | 158593 |
| Kansas | 1298163 | 1293371 | 1254278 | 1219387 | 1255779 | 1269960 | 1271563 | 1298644 | 1348197 | 1213904 |
| Kentucky | 24637 | 24648 | 27565 | 26845 | 23918 | 34820 | 35942 | 40236 | 37892 | 37695 |
| Louisiana | 457305 | 464418 | 461525 | 454552 | 474357 | 466391 | 480660 | 453071 | 456294 | 520158 |
| Maine | 10911 | 9281 | 11768 | 13246 | 12731 | 13232 | 14101 | 12322 | 12993 | 12004 |
| Maryland | 53551 | 64218 | 57960 | 57262 | 56885 | 65756 | 63208 | 64878 | 63111 | 62859 |
| Massachusetts | 5127 | 5277 | 5190 | 5125 | 4462 | 4867 | 4829 | 4676 | 4710 | 4883 |
| Michigan | 255967 | 249827 | 271441 | 280475 | 270287 | 298783 | 312146 | 319282 | 304834 | 307379 |
| Minnesota | 246865 | 248258 | 259921 | 265191 | 265084 | 261758 | 276892 | 284753 | 289736 | 268822 |
| Mississippi | 581096 | 592122 | 586089 | 574756 | 661108 | 607189 | 623127 | 586732 | 607843 | 727048 |
| Missouri | 632621 | 677098 | 639719 | 630628 | 594163 | 663467 | 689456 | 714821 | 700031 | 757763 |
| Montana | 755680 | 751866 | 765259 | 749722 | 709597 | 767409 | 752198 | 749607 | 734882 | 717839 |
| Nebraska | 3809427 | 3795113 | 3812085 | 3906419 | 3599322 | 3788075 | 3890195 | 3938095 | 3916648 | 3932941 |
| Nevada | 254701 | 253634 | 257118 | 254508 | 242259 | 255488 | 260470 | 263629 | 258504 | 261773 |
| New Hampshire | 1136 | 979 | 998 | 1051 | 1052 | 1033 | 1086 | 989 | 824 | 841 |
| New Jersey | 31874 | 32858 | 29627 | 32556 | 29711 | 29231 | 31729 | 30367 | 30056 | 27307 |
| New Mexico | 302674 | 309613 | 322031 | 287084 | 267775 | 303436 | 309672 | 315822 | 314231 | 278521 |
| New York | 22813 | 22985 | 21997 | 21375 | 19613 | 22087 | 21278 | 23348 | 21348 | 18062 |
| North Carolina | 39903 | 44633 | 40770 | 48174 | 49348 | 54476 | 60055 | 61742 | 62867 | 60946 |
| North Dakota | 192548 | 216921 | 222074 | 219590 | 208126 | 200636 | 237311 | 227049 | 208587 | 194485 |
| Ohio | 12389 | 16820 | 15163 | 17568 | 16019 | 20830 | 19226 | 21596 | 20769 | 21797 |
| Oklahoma | 241465 | 237963 | 255306 | 207626 | 222411 | 254679 | 253346 | 267197 | 280375 | 262775 |
| Oregon | 679313 | 689700 | 683527 | 672638 | 601796 | 681881 | 686693 | 687199 | 693521 | 630691 |
| Pennsylvania | 7463 | 5875 | 5829 | 5862 | 5100 | 5071 | 4593 | 4810 | 4193 | 3633 |
| Rhode Island | 1055 | 899 | 873 | 996 | 903 | 931 | 947 | 973 | 907 | 825 |
| South Carolina | 62819 | 60304 | 60997 | 70816 | 75314 | 78853 | 79582 | 78073 | 83985 | 83152 |
| South Dakota | 282989 | 300158 | 316604 | 305400 | 247765 | 310061 | 311233 | 310428 | 306264 | 273417 |
| Tennessee | 47716 | 50000 | 62366 | 72275 | 82320 | 95120 | 99624 | 105668 | 98550 | 95929 |
| Texas | 1947439 | 1958584 | 2042729 | 1806539 | 1734962 | 1950095 | 1931265 | 1990790 | 1960109 | 1797103 |
| Utah | 410925 | 413886 | 410487 | 415526 | 408438 | 409979 | 411089 | 407372 | 409971 | 404194 |
| Vermont | 1343 | 1497 | 1467 | 1250 | 1429 | 1135 | 1024 | 1176 | 953 | 988 |
| Virginia | 29632 | 30258 | 26965 | 30214 | 28495 | 31826 | 31587 | 32060 | 31459 | 30626 |
| Washington | 684029 | 689341 | 683064 | 675888 | 643850 | 688835 | 693133 | 687186 | 681781 | 649433 |
| West Virginia | 532 | 521 | 413 | 562 | 291 | 511 | 614 | 462 | 381 | 515 |
| Wisconsin | 197534 | 200655 | 213834 | 214935 | 208410 | 206657 | 216340 | 220279 | 222335 | 220264 |
| Wyoming | 554241 | 556873 | 553396 | 552743 | 506874 | 553539 | 554765 | 555745 | 552047 | 535443 |



| CONUS | 23902407 | 24082816 | 24182969 | 23875893 | 23064913 | 24180935 | 24400166 | 24660482 | 24579564 | 24185708 |
|---|---|---|---|---|---|---|---|---|---|---|

## Author contributions

Yanhua Xie: conceptualization, method design, result analysis, original draft writing, and manuscript review editing; Holly Gibbs: manuscript review and editing; Tyler Lark: conceptualization, funding acquisition, and manuscript review and editing. All authors have reviewed and agreed on the published version of the manuscript.

## Competing interests

The authors declare that they have no conflict of interest.

## Acknowledgement

This research was supported by the United States Geological Survey (Award Number: G19AC00080) and the Great Lakes Bioenergy Research Center, U.S. Department of Energy, Office of Science, Office of Biological and Environmental Research (Award Number: DE-SC0018409). The authors would like to thank members of the USGS Water Budget and Estimation Project for sharing verified irrigation data, feedback and ideas for mapping, and insights regarding data use for water estimation. We also appreciate the comments from anonymous reviewers and editors for their helpful suggestions. Any use of trade, firm, or product names is for descriptive purposes only and does not imply endorsement by the U.S. Government.

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
