# Peer review of "Landsat-based Irrigation Dataset (LANID): 30-m resolution maps of irrigation distribution, frequency, and change for the U.S., 1997-2017"

_Earth System Science Data, 2021_

## Author Comment (AC1)

The manuscript by Xie et al. "Landsat-based Irrigation Dataset (LANID): 30-m resolution maps of irrigation distribution, frequency, and change for the U.S." developed new irrigation mapping datasets in the US for both cropland and pasture with high spatial resolution across a relatively long-time span (1997-2017). The irrigation mapping showed high accuracy compared with validation, and other multiyear results revealed interesting regional and local patterns in irrigation changes. This work will be an important contribution to the community. The manuscript is well-written and the presentation is clear.

*** Response: Thank you for your comments, suggestions, and perspectives on our paper. We greatly appreciate the insights you have provided, and have revised the manuscript (highlighted in yellow) based on your comments accordingly, as detailed below.*

Below I have some minor comments:

L65. There are several different irrigation products used in this study. For readers who are am not familiar with each of them like me, it is helpful to add a column in the table to briefly summarize the method for producing each dataset.

*** Response: Thanks for the suggestion. We have added the methods used for each product in Table 1. A copy of the extended table is also provided here (method column is highlighted).*

Table 1. Currently available irrigation maps covering part to the entire CONUS. The boldfaced maps are compared with LANID in the Results section. (RF: random forest; RS: remote sensing)

| Products | Spatial coverage | Resolution | Update frequency | Methods/datasets | Citations |
|---|---|---|---|---|---|
| **Global Irrigated Area Map (GIAM)** | Global | 10 km rescaled to 1 km | Single map, 2000 | Spectral matching/RS data | Thenkabail et al. (2009) |
| **Global Map of Irrigation Areas (GMIA)** | Global | 10 km | 5-year, 1995, 2000, and 2005 | Spatial allocation/sub-nation statistics & maps | Siebert et al. (2005); Siebert et al. (2013) |
| Synthesized map of global irrigated area | Global | 1 km | Single map, covering 1999-2012 | Decision tree/RS, GMIA, & land cover maps | Meier et al. (2018) |
| Global Food-Support Analysis Data (GFSAD) | Global | 1 km | Single map, 2010 | Spectral matching/RS time series | Teluguntla et al. (2015) |
| Global Land Cover Map (GlobCover) | Global | 300 m | Single map, 2009 | Automatic classification/ RS time series | ESA (2015) |
| Global Land Cover Characteristics (GLCC) | Global | 1 km | Single map, 1992 | Hybrid compositing techniques/RS data | Loveland et al. (2000) |
| Global Rainfed, Irrigated and Paddy Croplands (GRIPC) | Global | 500 m | Single map, 2005 | Decision tree/RS, climate, & ag. inventory data | Salmon et al. (2015) |
| **MODIS-based Irrigated Agriculture Dataset (MIrAD)** | CONUS | 250 m | 5-year interval, 2002-2017 | Thresholding/ag. Census & RS data | Pervez and Brown (2010) |
| **MODIS-based Irrigation Fraction (MIF)** | CONUS | 500 m | Single map, 2001 | Decision tree/RS time series | Ozdogan and Gutman (2008) |

| | | | | | |
|---|---|---|---|---|---|
| **USDA-NASS Irrigation Statistics** | U.S. | County-level | 5-year interval, 1997-2017 | Surveys | https://www.nass.usda.gov/AgCensus/index.php |
| USGS-verified irrigated lands | Western U.S. | Field | Vary across states, 2002-2017 | Visual interpretation/RS & cropland inventory data | Brandt et al. (2021) |
| Landsat-based Irrigation Dataset 2012 (LANID 2012) | CONUS | 30-m | Single map, circa 2012 | RF/RS, climate, & envi data | Xie et al. (2019) |
| **Annual Irrigation Maps – High Plain Aquifer (AIM-HPA)** | High Plains Aquifer | 30-m | Annual, 1984-2017 | RF/RS, climate, & envi data | Deines et al. (2019) |
| **IrrMapper** | Western CONUS | 30-m | Annual, 1986-2018 | RF/RS, climate, & envi data | Ketchum et al. (2020) |

L125: What years are those selected reference and validation points?

*\*\* Response: We collected approximately 10,000 samples across the eastern US. Each irrigation sample records a center pivot location and the presence of irrigation infrastructure during the period of 1997-2017; each rainfed sample is a non-irrigated location for each year from 1997 to 2017. We stated the attributes of collected samples in the Section 4.1.*

L264: The authors need to discuss the spatial scale issue when comparing different datasets (points vs. pixel, spatial resolutions). How do different spatial resolutions influence the comparison among different data sources? What is a fair comparison? For example, how to make a fair comparison between irrigation fraction and the binary irrigation map?

*\*\* Response: Thanks for the insights. Given different spatial resolutions of existing maps, we used two ways to compare them: 30 m resolution, pixel-by-pixel comparison using ground truth data (Table 4) and subpixel comparisons (Figure 15). For the pixel-by-pixel comparison, we rasterized the ground truth data to 30 m resolution pixels and overlaid them with our maps and the existing binary ones (i.e., MIrAD, GIAM, AIM-HPA, IrrMapper, and LANID2012) to calculate accuracy metrics shown in Table 4. Because all binary maps tend to show exact locations of irrigated and non-irrigated croplands, we believe this pixel-by-pixel comparison can evaluate the locational accuracy of these maps. We also compared our maps with existing coarser resolution maps (i.e., MIrAD, GIAM, and GMIA) through a subpixel analysis shown in Figure 15. To do this, we aggregated our 30-m LANID maps to match the spatial resolution of each product (e.g., 250-m for MIrAD and 10-km for GMIA). This subpixel comparison shows irrigation fraction of mapped irrigation locations on each binary product (MIrAD and GIAM) and how well fraction products (MIF and GMIA) estimate irrigation proportion within coarse resolution pixels.*

L296 and Fig 14b: Unlike other products which were only compared with one year of NASS data, LANID was compared with multiple years of NASS data. I think this may contribute to the higher R2 of LANID in Fig. 14b. In terms of R2, LANID is better, but I don't understand why it is written that the LANID agreement is weaker than MirAD and GMIA.

*\*\* Response: We stated that our LANID performs better than MIF and GIAM at the state level but worse than MIrAD and GMIA because these later two products used USDA-NASS reported*

*area as the reference to downscale to the pixel scale. Given this, MIrAD and GMIA match perfectly with the USDA-NASS reported amount from county to country scales (also showed in Figure 14a at the country scale). To reduce confusion, we added methods each product used in Table 1, which also echoes to the reviewer's first request of adding method summary.*

*We plotted all five-year data in the same plot in Figure 14b because each year follows a similar pattern. However, we believe it is not difficult to tell that our data shows higher consistency with USDA-NASS reported areas (along the 1:1 line with no states showing substantial overestimation or underestimation) than MIF and GIAM, which show substantially underestimation and overestimation, respectively.*

---

## Author Comment (AC2)

This is a well-written paper describing the methodology used to produced CONUS-wide, 30 m resolution maps of annual irrigation status. This paper effectively presents results from analysis showing the time series of irrigated area state-by-state, irrigated change at the county and state level, and maps change spatially over the CONUS. Figures are clear and easy to read.

*** Response: Thank you for your comments and your time invested on this paper. We have responded to each comment (except for some positive ones) and revised the manuscript accordingly.*

Abstract is appropriately specific and clearly states the need for this product, general methodological approach, and utility of the produced data.

-- The link to data seems to be the data published with the RSE manuscript, and appears not to contain maps showing the new LANID_V2 data mapping irrigated hay and pasture. Should this be updated, or another repository offered for the examination of the new data described in this manuscript?

*** Response: Thanks for your comment on data link, which might be misleading because we set our RSE paper as the preview option. We did this because this RSE paper describes the detailed methods used to create Version 1.0 LANID (i.e., irrigated pasture and hay were not included). The data link provided in the Abstract and Data Availability Sections of this paper is definitely for the Version 2.0 LANID, which covers all maps (i.e., annual irrigation extent maps, irrigation frequency, and change) and our collected ground reference data. We do have the Version 1.0 LANID, but its link was only provided in the RSE paper (Google Earth Engine Asset Id: "users/xyhuwmir4/LANID/LANID_v1_rse").*

*Upon publication of the current ESSD dataset and paper, we will update the Zenodo repository accordingly to ensure clarity and to include the ESSD manuscript to reduce confusion.*

Introduction is well-written and provides a good summary of why irrigation is important, and the impacts and benefits of irrigation. Literature review is appropriately specific and comprehensive. Table 1 is comprehensive and appropriate.

Methods section is clear and concise, and describes a sound approach given the challenge of detecting irrigation from Landsat images. The doption of two different approaches for detection in the humid East and semi-arid West US, while adding complexity, is justified given the low contrast between irrigated and non-irrigated lands in the East.

Map evaluation and comparison designs seems to choose appropriate, previously produced maps for comparison to LANID.

Figure 4 is epecially attractive. Should Figure 4 take into account uncertainty estimates?

*** Response: Thanks for this consideration. Figure 4 shows LANID-derived temporal trends of irrigation area per state. As we did not have sufficient ground reference data to evaluate map accuracies per state (instead by regions in this manuscript – West, NKOT, and East), it is not currently possible for us to provide uncertainty estimates by state. We will provide such*

*information in our future versions of LANID when we have more ground reference data, especially for the Mississippi Alluvial Plain region.*

*Given clear pattens shown in Figure 4, we did not conduct trend analysis (e.g., linear regression) and associated trend uncertainties. For readers who want to investigate more, please refer to the time-series state-level irrigation area provided in Table A1.*

Figures 5, 6, and 7 are informative and well done.

Irrigated pasture and hay: where is this data in the repo referred to in the abstract?

*\*\* Response: The annual maps under the link include both irrigated croplands and pasture/hay, so the thematic maps of irrigated pasture and hay can be easily created by overlaying our LANID maps and "pasture/hay" classes from publicly available USGS National Land Cover Database and USDA Cropland Data Layers, like the maps showed in Figure 8. To be more convenient for users, we have updated the Zenodo repository to include this layer as "irrFreqPasture_West.tif". The DOI of the new version is https://doi.org/10.5281/zenodo.5548555, which is also updated in the manuscript (the old one still works).*

Maximum extent, frequency, and formerly irrigated and intermittent irrigated land: interesting findings. Line 259: what is meant by 'energy sorghum'? Table 3 is interesting and informative.

*\*\* Response: Thanks for pointing out this need for clarification! There is no formal definition for energy sorghum, but the term generally refers to those varieties that are high-yielding, photoperiod sensitive, and potentially suitable as bioenergy feedstocks, as explained by Cui et al. 2018. We have added citations to this and two additional articles that help clarify and further describe this concept.*

*Cui, X., Kavvada, O., Huntington, T., & Scown, C. D. (2018). Strategies for near-term scale-up of cellulosic biofuel production using sorghum and crop residues in the US. Environmental Research Letters, 13(12), 124002.*

*Mullet, J., Morishige, D., McCormick, R., Truong, S., Hilley, J., McKinley, B., ... & Rooney, W. (2014). Energy Sorghum—a genetic model for the design of C4 grass bioenergy crops. Journal of experimental botany, 65(13), 3479-3489.*

*Enciso, J., Jifon, J., Ribera, L., Zapata, S. D., & Ganjegunte, G. K. (2015). Yield, water use efficiency and economic analysis of energy sorghum in South Texas. Biomass and Bioenergy, 81, 339-344.*

Figures 11 - 13 clearly display the improvement in mapping resolution and accuracy over previous maps.

The comparison of irrigated area maps is clear and offers cogent explanations of why differences in the maps exist, in terms of differences in irrigated lands' definition and classification methodology.

The discussion of uncertainty, limitations, improvements and potential applications is appropriate.